# Projected impact of COVID-19 mitigation strategies on hospital services in the Mexico City Metropolitan Area

Zachary Fowler[1]*, Ellie Moeller[1,2], Lina Roa[1,3], Isaac Deneb Castañeda-Alcántara[4], Tarsicio Uribe-Leitz[1,5,6], John G. Meara[1,7], Arturo Cervantes-Trejo[4]

1 Program in Global Surgery and Social Change, Harvard Medical School, Boston, Massachusetts, United States of America, 2 University of Miami Miller School of Medicine, Miami, Florida, United States of America, 3 Department of Obstetrics & Gynecology, University of Alberta, Edmonton, Alberta, Canada, 4 Anahuac Institute of Public Health, Faculty of Health Sciences, Anahuac University Mexico, Huixquilucan, State of Mexico, Mexico, 5 Center for Surgery and Public Health, Brigham and Women's Hospital, Boston, Massachusetts, United States of America, 6 Department of Epidemiology, Technical of University Munich, Munich, Bavaria, Germany, 7 Department of Plastic and Oral Surgery, Boston Children's Hospital, Boston, Massachusetts, United States of America

* zachary.g.fowler@gmail.com

**Data Availability Statement:** All relevant data are within the manuscript and its Supporting information files.

## Abstract

Evidence-based models may assist Mexican government officials and health authorities in determining the safest plans to respond to the coronavirus disease 2019 (COVID-19) pandemic in the most-affected region of the country, the Mexico City Metropolitan Area. This study aims to present the potential impacts of COVID-19 in this region and to model possible benefits of mitigation efforts. The COVID-19 Hospital Impact Model for Epidemics was used to estimate the probable evolution of COVID-19 in three scenarios: (i) no social distancing, (ii) social distancing in place at 50% effectiveness, and (iii) social distancing in place at 60% effectiveness. Projections of the number of inpatient hospitalizations, intensive care unit admissions, and patients requiring ventilators were made for each scenario. Using the model described, it was predicted that peak case volume at 0% mitigation was to occur on April 30, 2020 at 11,553,566 infected individuals. Peak case volume at 50% mitigation was predicted to occur on June 1, 2020 with 5,970,093 infected individuals and on June 21, 2020 for 60% mitigation with 4,128,574 infected individuals. Occupancy rates in hospitals during peak periods at 0%, 50%, and 60% mitigation would be 875.9%, 322.8%, and 203.5%, respectively, when all inpatient beds are included. Under these scenarios, peak daily hospital admissions would be 40,438, 13,820, and 8,650. Additionally, 60% mitigation would result in a decrease in peak intensive care beds from 94,706 to 23,116 beds and a decrease in peak ventilator need from 67,889 to 17,087 units. Mitigating the spread of COVID-19 through social distancing could have a dramatic impact on reducing the number of infected people and minimize hospital overcrowding. These evidence-based models may enable careful resource utilization and encourage targeted public health responses.

**Funding:** The authors received no specific funding for this work.

**Competing interests:** The authors have declared that no competing interests exist.

## Introduction

The outbreak of the novel coronavirus disease 2019 (COVID-19), first identified in the Hubei Province in China in December 2019, was declared a Public Health Emergency of International Concern by the World Health Organization (WHO) on January 30, 2020 [1, 2]. Having now infected more than 40 million people in nearly every country, the COVID-19 pandemic has resulted in more than 1.1 million reported deaths globally [3]. Many affected countries have implemented mitigation measures, such as social distancing, quarantine, and contact tracing to minimize viral spread and death [4].

In Mexico as of October 21, 2020, there have been approximately 860,000 confirmed cases of COVID-19 and more than 86,000 reported deaths, although some experts have estimated the true values to be up to 50 times greater than those reported [5, 6]. This underestimation is likely due in part to the lack of COVID-19 testing, with Mexico performing the fewest tests of any of the Organization for Economic Cooperation and Development (OECD) and Latin American countries with available data. The first case was reported on February 28, 2020, and as of May 5, 2020, Mexico was performing 0.69 tests per 1,000 people, which is 55 times lower than the average OECD country [7, 8]. Of people screened, 44% have tested positive, and the country has a case fatality rate of 10.56%–substantially higher than the global case fatality rate of 3.12% [9]. Among patients requiring mechanical ventilation, mortality in Mexico has also been shown to be significantly higher than countries such as the United Kingdom and Germany [10]. According to a recent technical report by the Scientific Technical Commission analyzing COVID-19 deaths, there was an excess mortality of 22,366 deaths (143% more than expected) in Mexico City from April to June 30, 2020. This excess was greatest in the population aged 45 to 60 years (278% excess) [11]. From January to August 2020, the Digital and Public Innovation Agency reported that Mexico City had an excess of 30.462 deaths, and more than 20,000 were directly associated with COVID-19 [12]. However, recent reports using death certificate registries in Mexico City have shown the number of deaths due to COVID-19 to be nearly three times higher than figures reported by the Ministry of Health [13].

With a significant rise in infections in the country, Mexico entered the government-declared "Phase 3" of COVID-19 on April 21, 2020, the most severe phase in which community transmission is widespread and cases are in the thousands [14]. Beginning in late March, Mexico implemented mitigation measures that include school closures, suspension of gatherings of more than 100 people, and limiting all non-essential activities [15, 16].

The majority of the confirmed cases and deaths have occurred in greater Mexico City, the financial, economic, and political center of the country where 21 million people reside [17–19]. Spreading over a large valley, the megalopolis is known officially as the Metropolitan Zone of the Valley of Mexico and hereon as the Mexico City Metropolitan Area (MCMA). It is one of the largest urban conglomerates in the world, consisting of Mexico City's 16 municipalities as well as 59 surrounding municipalities in the State of Mexico and 1 municipality (Tizayuca) in the State of Hidalgo. In this urban conglomerate that spans 7,866 square kilometers, there are 3,535 registered healthcare units in the public and private sectors [20].

Despite global evidence pointing to the utility of implementing such mitigation measures, there have yet to be any evidence-based predictions of the potential impact of these policies in the MCMA. Model-based predictions can serve as a valuable tool for health authorities and policy makers to construct an informed response to the pandemic [4]. Many models have now been developed, offering a wide range of predictions which include hospital beds, intensive care unit (ICU) capacity, ventilators, physician and other staff needs, and PPE consumption [21–23]. The input variables differ among the various models and include factors such as local hospital admission rates, case severity, clinical care pathways, population age distribution,

asymptomatic transmission rate, and unreported death rates [21, 24, 25]. Various methodologies, including simulation and compartmental models with deterministic and stochastic systems, have been proposed, and artificial intelligence and machine learning tools have been employed to guide interventions [21, 26].

In Mexico, despite the availability of prediction models that can be applied to a variety of needs and setting, previous model-based predictions have not included possible mitigation strategies and have not been focused on the most affected area of the MCMA, thereby limiting the ability to guide public health policy [27, 28]. An analysis of the MCMA is critical to estimate possible transmission patterns that may be observed in the metropolitan area where 17% of the nation's population resides [18]. Evidence-based models are necessary to help Mexican government officials and health authorities determine the safest plans to combat the COVID-19 pandemic in the most-affected region. As such, the purpose of this study is to present the potential impacts of COVID-19 in the MCMA and to better understand the consequences of its mitigation efforts.

## Methods

### Model parameters

To make an illustrative, predictive estimate of the hospital impact for MCMA hospitals during the COVID-19 pandemic, we used the COVID-19 Hospital Impact Model for Epidemics (CHIME) [29]. This provides a deterministic model on the probable evolution of the disease with and without mitigation interventions. The severity was considered from a set of demographic, epidemiological and capacity variables of the healthcare system. CHIME was recently developed by the University of Pennsylvania to assist hospitals and public health officials with capacity planning in the United States during the COVID-19 pandemic [29].

The projections calculated by CHIME are based on the "Susceptible, Infected and Recovered" (SIR) model to estimate the population impact. SIR modeling is a technique applied in epidemiology to estimate the number of susceptible ($S$), infected ($I$), or recovered ($R$) individuals in a given population and over a certain period of time [29]. The model considers that the COVID-19 pandemic will continue through a process of growth and decline, according to the characteristics of infectious disease spread. CHIME provides estimates of total new and running inpatient hospitalizations, ICU admissions, and patients requiring mechanical ventilation. The projections for hospital occupancy and the use of ICU beds and ventilators are based on the clinical parameters observed in COVID-19 patients in the United States from the American Hospital Association and Penn Medicine facilities.

The model calculates the basic reproduction number (denoted $R_0$), which is the number of infections that are expected to occur as a result of one case, assuming that the entire population is susceptible [30]. The model also calculates the effective reproduction rate (denoted $R_t$) which indicates the number of cases anticipated under the conditions at a specific point in time [31].

Epidemic characteristics and healthcare needs are calculated using two parameters: effective contact rate ($\beta$) and the inverse mean recovery time ($\gamma$). Effective contact rate is the product of transmissibility and the average number of individuals exposed. Transmissibility is a measure of the virulence of the virus, and the number of people exposed can be modified with social distancing [29].

The model uses a value for $\gamma$ of 1/10, which reflects the recommendation by the United States Centers for Disease Control and Prevention to self-quarantine for 10 days following a known exposure [32]. The value for $\beta$ is extracted from known doubling times ($T_d$). The

growth rate ($g$) can be calculated with the following formula [29]:

$$g = 2^{1/Td} - 1$$

β can be determined as the sum of $g$ and $\gamma$, and the SIR dynamics are then determined as follows [29]:

$$S_{t+1} = S_t - \beta S_t I_t$$

$$I_{t+1} = I_t + \beta S_t I_t - \gamma I_t$$

$$R_{t+1} = R_t + \gamma I_t$$

## Scenarios

Three scenarios were run in the model for the MCMA: (i) no social distancing, (ii) social distancing implemented at 50% effectiveness, and (iii) social distancing implemented at 60% effectiveness, as a best-case scenario. These values were chosen for illustrative purposes under three hypothetical situations. It was estimated that in China, mitigation of 50–60% would be needed to reverse the epidemic, and 60% has been used in recent prediction studies evaluating maximum reduction strategies [33–35]. This method of using 0%, 50%, and 60% allows us to observe the differences in service demand under incremental scenarios and understand the graduality of the impact of mitigation measures.

## Model inputs

Model inputs include regional population, hospital market share, number of patients currently hospitalized with COVID-19, date of the first hospitalization, social distancing, hospitalization rate, ICU admission rate, mechanical ventilation rate, infectious days, average length of hospital stay, average length of ICU stay, and average length of ventilator use [29].

The initial projections were based on correspondence of the approximate number of hospitalizations on April 2, 2020 in the MCMA with inputs listed in Table 1. This was repeated on May 7, 2020 when there were 4,406 reported hospitalizations in the MCMA [36]. The remaining inputs were unchanged in the first and second models. Hospitalization rate (2.5%), ICU admission rate (0.75%), and mechanical ventilation rate (0.5%) were based on estimates by Verity *et al* using data from mainland China and 37 other countries in January and February 2020 [37]. Inputs for average hospital length of stay, average ICU length of stay, and average number of days requiring mechanical ventilation were based on data from patients with respiratory failure treated at four Penn Medicine facilities over a five year period. Hospital market share is defined as the percentage of patients who are likely to present to the hospitals included in the analysis, and it is estimated by the number of hospital beds included in the study compared to the total number of hospital beds in the region [29].

## Data

The population used for the MCMA corresponds to estimates by the National Urban System and published by the National Population Council 2018 [38]. The data on health infrastructure, including the number of secondary and tertiary care units, hospital in-patient beds, total beds, number of doctors and nurses, and accreditation were obtained from a publicly available database that was most recently updated in 2015 by the federal government [39].

**Table 1. Inputs used for CHIME projections modeled on April 2 and May 7, 2020.**

| | |
|---|---:|
| Mexico City Metropolitan Area population | 21,800,322 |
| Percentage of secondary and tertiary hospitals included (n = 811) | 100% |
| Patients currently hospitalized with COVID-19 (as of April 2, 2020) | 1,000 |
| Patients currently hospitalized with COVID-19 (as of May 7, 2020) | 4,406 |
| Date of first hospitalization due to COVID-19 (as reported officially) | March 9, 2020 |
| Social distancing worst case scenario | 0% |
| Social distancing intermediate case scenario | 50% |
| Social distancing best case scenario | 60% |
| Date of social distancing put in place | 24-Mar-20 |
| Percentage of infected patients requiring hospitalization | 2.50% |
| Percentage of infected patients requiring intensive care | 0.75% |
| Percentage of infected patients requiring mechanical ventilation | 0.50% |
| Infectious period | 10 days |
| Duration of hospitalization | 7 days |
| Duration of intensive care (includes patients without need for mechanical ventilation) | 9 days |
| Duration of mechanical ventilation | 10 days |

## Healthcare facilities

According to the most recent official data, the MCMA has 822 secondary and tertiary healthcare units in the private and public sectors. These health units have 29,569 hospitalization beds, 14,357 doctors, and 22,611 nurses. Few of these units are accredited by the corresponding specialty councils or hospital accreditation authorities of the country (Table 2). Of the 3,534 registered healthcare units in the MCMA, 77% are primary care, 21% are secondary care, and only 2% are tertiary care hospitals [19].

Table 3 displays how these general (secondary) and specialty (tertiary) hospitals are distributed by provider type. 75.3% of all secondary and tertiary hospitals in the MCMA belong to the private sector. In the public sector the largest provider of second and third level hospitals is the Ministry of Health of the Federal Government, followed by the Mexican Institute of Social Security, the Institute for Social Security and Services for State Workers, and military hospitals.

## Results

### No social distancing

With no mitigation efforts through social distancing, peak case volume was predicted to occur on April 30, 2020 at 11,553,566 infected individuals. This is based on inputs listed in Table 1, which results in an initial doubling time of 1.998 days and a recovery time of 10 days. It implies a $R_0$ of 10 and daily growth rate of 41.46%. The absence of social contact reduction after the onset of the outbreak results in a $R_t$ of 5.15 and daily growth rate of 41.46%.

The highest need for hospital beds was predicted to occur on April 28, 2020 at 258,999 beds. For critically ill patients, the maximum ICU occupancy was predicted to occur on April 29, 2020 with a need for 94,706 ICU beds. The highest need for mechanical ventilators to support critically ill patients was predicted to occur on April 29, 2020 with 67,889 ventilators needed.

### Social distancing effective at 50%

With mitigation efforts through social distancing effective at 50%, peak case volume was predicted to occur on June 1, 2020 with 5,970,093 infected individuals. The highest need for

**Table 2. Number of healthcare units and key indicators in the MCMA.**

| Geographic Location | Healthcare Units | Hospitalization Beds | Total Beds* | Doctors⁋ | Nurses⁋ | Accredited Units‡ |
|---|---|---|---|---|---|---|
| **Mexico City** | **639** | **21,731** | **22,618** | **9,515** | **14,574** | **55 (8.6%)** |
| Secondary Care Units | 579 | 12,096 | 12,675 | 2,315 | 3,240 | 21 (3.6%) |
| Tertiary Care Units | 60 | 9,635 | 9,943 | 7,200 | 11,334 | 34 (56.7%) |
| **State of Mexico** | **172** | **7,787** | **8,957** | **4,842** | **8,037** | **30 (17.4%)** |
| Secondary Care Units | 161 | 6,429 | 7,320 | 3,511 | 7,098 | 23 (14.3%) |
| Tertiary Care Units | 11 | 1,358 | 1,637 | 1,331 | 939 | 7 (63.6%) |
| **State of Hidalgo** | **31** | **51** | **51** | **N/A** | **N/A** | **0 (0%)** |
| Secondary Care Units | 11 | 51 | 51 | N/A | N/A | 0 (0%) |
| Tertiary Care Units | - | - | - | - | - | - |
| **Grand Total** | **822** | **29,569** | **31,626** | **14,357** | **22,611** | **85 (10.4%)** |

*Total beds includes hospitalization beds and non-hospitalization beds.

⁋ Includes only public sector doctors and nurses in contact with patients. Excludes psychiatric hospitals.

‡ Accredited by the national accreditation bodies of distinct specialties. N/A = data not available.

**Table 3. Healthcare units in the MCMA.**

| State (number of municipalities) | Secondary Care | Tertiary Care | Total |
|---|---|---|---|
| **Mexico City (16)** | **579** | **60** | **639** |
| Mexican red cross | 1 | - | 1 |
| ISSSTE | 15 | 4 | 19 |
| IMSS | 28 | 9 | 37 |
| Petroleos mexicanos | - | 2 | 2 |
| Secretary of defense | 6 | 2 | 8 |
| Secretary of navy | - | 2 | 2 |
| Ministry of health | 23 | 37 | 60 |
| State medical services | 1 | - | 1 |
| Private medical services | 505 | 4 | 509 |
| **State of Mexico (59)** | **161** | **11** | **172** |
| ISSSTE | 1 | 1 | 2 |
| IMSS | 17 | 1 | 18 |
| Secretary of defense | 7 | - | 7 |
| Ministry of health | 30 | 8 | 38 |
| State medical services | 6 | 1 | 7 |
| Private medical services | 100 | - | 100 |
| **State of Hidalgo (1)** | **11** | **0** | **11** |
| ISSSTE | - | - | - |
| IMSS | 1 | - | 1 |
| Ministry of health | - | - | - |
| Private medical services | 10 | - | 10 |
| **Total** | **751** | **71** | **822** |

IMSS = Mexican Institute of Social Security; ISSSTE = Institute for Social Security and Services for State Workers.

hospital beds was predicted to occur on May 27, 2020 at 95,448 beds. For critically ill patients, the maximum ICU occupancy was predicted to occur on May 27, 2020 with a need for 36,493 ICU beds. The highest need for mechanical ventilators to support critically ill patients was predicted to occur on May 28, 2020 with 26,899 ventilators needed.

The model was re-run on May 7, 2020 with 4,406 hospitalizations and social distancing was estimated at 50% effectiveness. Under this scenario, the estimated number of infected individuals on May 7, 2020 was 180,066. An initial doubling time of 2.57 days and a recovery time of 10 days imply a $R_0$ of 4.10 and daily growth rate of 30.99%. With a 50% reduction in social contact after the onset of the outbreak the doubling time is reduced to 6.9 days. This leads to a $R_t$ of 2.05 and daily growth rate of 10.49%.

## Social distancing effective at 60%

With mitigation efforts through social distancing effective at 60%, peak case volume was predicted to occur on June 21, 2020 at 4,128,574 infected individuals. The highest need for hospital beds was predicted to occur on June 16, 2020 at 60,176 beds. For critically ill patients, the maximum ICU occupancy was predicted to occur on June 16, 2020 with a need for 23,116 ICU beds. The highest need for mechanical ventilators to support critically ill patients was predicted to occur on June 16, 2020 with 17,087 ventilators needed.

The model was re-run on May 7, 2020 with 4,406 hospitalizations, and social distancing was estimated at 60% effectiveness. Under this scenario, the estimated number of infected individuals on May 7, 2020 was 184,359. An initial doubling time of 2.19 and a recovery time of 10 days imply a $R_0$ of 4.71 and daily growth rate of 37.14%. With a 60% reduction in social contact after the onset of the outbreak the doubling time is reduced to 8.2 days. This leads to a $R_t$ of 1.89 and daily growth rate of 8.85%.

## Resource utilization

In the MCMA there are 29,569 inpatient hospital beds in the public and private sectors at 817 secondary and tertiary hospitals of the MCMA. Occupancy rates during peak periods at 0%, 50%, and 60% mitigation would be 875.9%, 322.8%, and 203.5%, respectively, when all inpatient beds are considered. During these periods the percentage of the population infected was predicted to be 53.0% (at 0% mitigation), 27.4% (at 50% mitigation), and 18.9% (at 60% mitigation). Fig 1 demonstrates the projected number of cases (persons with COVID-19 infection), hospital bed needs, ICU bed needs, and ventilator needs over time at 0%, 50%, and 60% mitigation. Fig 2 shows the number of susceptible, infected, and recovered individuals over time at 0% and 60% mitigation scenarios (50% mitigation scenario was left out for clarity). Projections of peak daily infrastructure demand, overall infrastructure utilization, and susceptible, infected, and recovered individuals at 0%, 50%, and 60% mitigation are listed in Table 4.

## Discussion

This study illustrates how adequately enforced mitigation measures could reduce the number of individuals who are expected to become infected in the MCMA, from more than 11 million to approximately 6 million with mitigation measures effective at 50%. With further mitigation efforts, effective at 60%, the number of individuals who are expected to become infected could be further reduced to approximately 4 million individuals. Effective mitigation measures could also reduce the number of individuals who require hospitalization. This could be reduced from a high of 258,999 thousand hospitalizations during the peak period without any mitigation measures to 95,448 at 50% or 60,176 at 60% mitigation effectiveness. These results indicate that even under the best-case scenario (60% effectiveness in social distancing), the total

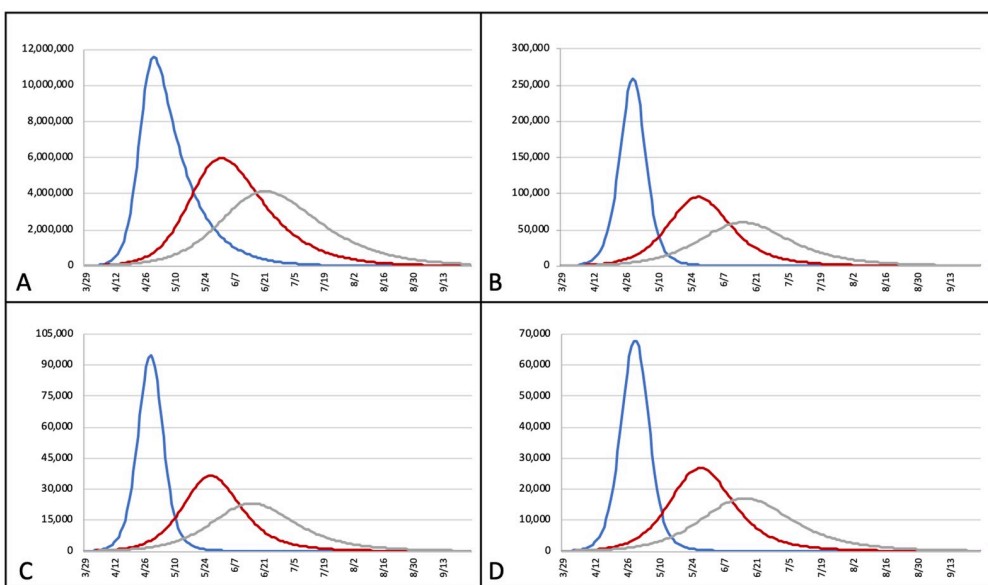

**Fig 1. Projected number of active cases (A), hospital bed occupancy (B), ICU bed occupancy (C), and ventilator utilization (D) by date with 0% (blue), 50% (red), and 60% (grey) mitigation scenarios.** X-axis = date; Y-axis = number of infected individuals (A), hospital beds (B), ICU beds (C), and ventilators (D).

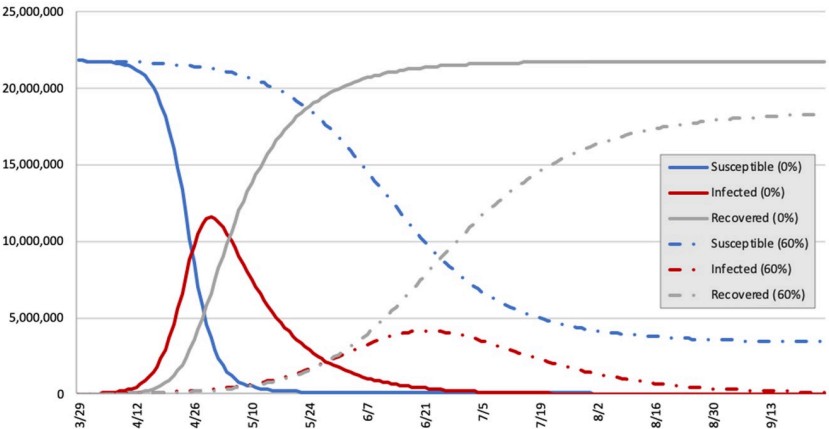

**Fig 2. Comparison of susceptible, infected, and recovered individuals in the MCMA with 0% and 60% mitigation through social distancing.** X-axis = date; Y-axis = number of individuals.

hospital bed capacity of 29,569 in the MCMA will not be sufficient to meet the potential demand for services.

The estimated demand that COVID-19 creates for public and private sector hospital ser-vices in the MCMA under different mitigation scenarios is useful for understanding the impact that the pandemic has for healthcare service providers. It also quantifies the expected results of sustained mitigation efforts, which is quite notable. From these results it is clear that social dis-tancing can be very useful for reducing the expected demand for healthcare services. These estimates can also serve to ascertain the effectiveness of mitigation efforts by comparing actual results of disease spread in the MCMA to what could have been expected without mitigation

**Table 4. Projections of peak health system needs and susceptible, infected, and recovered individuals.**

| Peak Projections | 0% Mitigation | 50% Mitigation | 60% Mitigation |
|---|---|---|---|
| New daily hospital admissions | 40,438 | 13,820 | 8,650 |
| New daily ICU admissions | 12,131 | 4,146 | 2,595 |
| New daily ventilator needs | 8,088 | 2,764 | 1,730 |
| Total hospital bed occupancy | 258,999 | 95,448 | 60,176 |
| Total ICU bed occupancy | 94,706 | 36,493 | 23,116 |
| Total ventilator utilization | 67,889 | 26,899 | 17,087 |
| Susceptible individuals | 21,800,282 | 21,800,282 | 21,800,282 |
| Infected individuals | 11,553,566 | 5,970,093 | 4,128,574 |
| Recovered individuals | 21,717,060 | 20,136,105 | 18,411,125 |

or control measures. The scenario with mitigation measures functioning at 60% effectiveness predicted that there would be 4,406 hospitalizations on April 26 or 27, 2020. This number was reached on May 7, 2020, suggesting that the peak may be reached later than initially projected at 60% mitigation. It is important to note that trends in case volume may be affected by holidays and cultural events that involve changes in social interaction.

Given that 75% of secondary and tertiary hospitals in the MCMA belong to the private sector, these hospital services are not readily accessible to the uninsured population. To expand coverage on April 5, 2020, the federal government issued the "Hospital Reconversion Guidelines", which were designed to expand the capacity of health services by modifying physical facilities in existing hospitals to increase COVID-19 hospitalization areas. This strategy, which has a cost estimated at 2 billion Mexican Pesos (89.7 million United States Dollars), has expanded the capacity of MCMA hospitalization services. Additionally, the Ministry of Health signed a collaboration agreement with the National Association of Private Hospitals and the Mexican Hospital Consortium to make half of the beds at 146 private hospitals available to the federal government [40].

Adequately enforced mitigation measures could also reduce the estimated ICU bed demand from more than 94,706 needed in the case of no mitigation measures to approximately 36,493 at 50% mitigation effectiveness or approximately 23,116 at 60% effectiveness. In any of these scenarios ICU bed demand is expected to surpass availability by a wide margin. Finally, effective mitigation measures could reduce the need for ventilator support from more than 67,889 patients to approximately 26,899 at 50% or 17,087 at 60% effectiveness. In each of these scenarios, ventilator demand is also expected to surpass current availability. Local authorities estimated that between May 6 and 10, 2020, the pandemic would peak in Mexico City with a demand for 2,500 ventilators [41]. However, the peak had not been reached at that time, and hospital occupancy was continuing to increase with most secondary and tertiary hospitals reporting near saturation [42].

These results demonstrate that social distancing could have a dramatic impact on reducing the number of infected people in Mexico, which is consistent with other studies. A study predicting the vulnerability of fifty Mexican cities found that a 45% adherence to social distancing could reduce the number of infected people by up to 78.7% [43]. While modeled data are useful in planning for health system needs, other countries have begun to measure the true impact of social distancing measures using interrupted time series analyses. Iran and the United States observed a daily decrease in the trend of new cases after implementing social distancing policies. The doubling time in the United States was noted to increase from 3.3 days to 5.0 days at 14 days after policy implementation [44, 45]. Another study in the United States found a

decrease in growth rate of 9.1% after 16–20 days of social distancing measures. Without these measures the authors predicted that disease spread would have been 35 times greater [46].

This modeling exercise can be used to construct scenarios for the potential impact of COVID-19 on the Mexican population, as well as to develop strategic plans to manage the predicted case volume. There is a need across multiple areas of the health system for modeled projections and planning. Specifically, projections can impact PPE distribution, interruption of routine healthcare services (such as surgery and endoscopy), production and distribution of needed medical equipment (including ventilators), and repurposing existing infrastructure (*e. g.*, operating rooms, post-anesthesia care units, anesthesia machines, and recruitment of additional physicians) [47, 48]. The Mexican Association of General Surgery, for example, recommended postponing all elective surgery and endoscopy, and it has been estimated that 12-week cancellation rates for surgery were 41.8% for cancer surgery, 80.7% for benign surgery, and 25% for obstetric surgery [49, 50]. Extensive planning involving multiple disciplines is required to make these changes, and modeled projections can bolster these efforts. For example, transformation of operating room spaces to ICUs will require expertise in engineering for conversion of the environment from positive pressure to negative pressure to reduce the risk of infection spread [51]. Employing models can help guide these critical decisions and influence which supporting disciplines are needed. Ultimately, this may enable careful resource utilization and swift public health responses that protect the entire MCMA, for which data reporting and public health management are often approached individually by each of the three states within their respective administrative boundaries. Analyses and prediction models that include all municipalities in the metropolitan area can lead to more comprehensive and efficient policies. Additionally, the impact of these predicted case volumes on areas other than health service delivery must be considered, including medical education, budget planning, and social and economic needs of the population.

## Limitations

We recognize that these healthcare demand estimates have several limitations. First, the CHIME application was developed for use in the United States, and the epidemiological parameters of incidence, hospitalizations, ICU admissions, and ventilator needs are based on published data from the United States. These parameters may be less applicable to settings outside of the U.S. due to differences in demographic, social, and economic conditions that may influence the spread and behavior of COVID-19. However, given the lack of reliable public data and local pandemic parameters for Mexico or the MCMA, we believe that these estimates are a useful reference point. In the absence of models developed in countries of lower income classifications, predictions using the CHIME application may provide a vantage point for estimating disease outbreaks and information to inform public health planning and preparedness. In addition to several areas of the United States, this model has been applied to predict healthcare needs and guide policy in Australia and Sri Lanka [35, 52–56].

Long term projections must also be used with caution. Unpredictable factors can alter the course of COVID-19 and decrease the accuracy of early projections. As in other new and emerging diseases, there is a high level of uncertainty regarding the characteristics of infection, transmission, and effectiveness of social distancing measures. Another key factor that could limit the accuracy of predictive models is the reliability of available data on the number of infections, hospitalizations, and complications, such as ICU admissions and ventilator usage. Mexico has the lowest COVID testing rate of any OECD country, which likely leads to a large underreporting of positive cases [8]. Therefore, a case volume that appears to be much lower than predicted may be, in part, attributable to missed cases.

Finally, implementation of mitigation measures has been reported to be heterogeneous given the influence of social determinants such as mobility, working and living conditions, and economic inequality. Federal health and local mobility authorities in Mexico City have reported decreased mobility of pedestrians as well as public and private transportation ranging 50–60% [57]. However, there are also widespread reports of mass gatherings and intense social activity around markets and commercial areas in many communities. This could result in varying degrees of spread within the MCMA. More affluent neighborhoods and communities may be better able to conform to mitigation measures, while others may be more exposed due to their need to work, their use of public transportation, or their living conditions. This could result in unequal contagion and demand for healthcare services in different parts of the metropolis.

## Conclusions

As the death toll due to COVID-19 in Mexico continues to rise and healthcare resources are strained, there is an urgent need to mobilize and allocate resources as efficiently as possible. Although the true scope of the pandemic is yet to be characterized, several conclusions can be drawn from our models:

- The healthcare system of the MCMA will become heavily overburdened even with the most successful mitigation measures in place.

- The public health crisis caused by COVID-19 demonstrates the value of planning and pre-paredness in Mexico, as well as the value of collaboration with universities and the use of scientific tools such as CHIME [29].

- Although these projections contain uncertainty, our repeated models demonstrate that social distancing efforts can mitigate the detrimental impact of COVID-19 in the MCMA.

- These projections could inform governmental and health professionals in determining public health policies and distribution of scarce vital resources. This can assist the Mexican government in continuing to respond to the ongoing pandemic in Mexico.

## Supporting information

**S1 Dataset.**
(XLSX)

**S2 Dataset.**
(XLSX)

**S3 Dataset.**
(XLSX)

**S4 Dataset.**
(CSV)

**S5 Dataset.**
(CSV)

## Author Contributions

**Conceptualization:** Zachary Fowler, Lina Roa, Tarsicio Uribe-Leitz, Arturo Cervantes-Trejo.

**Data curation:** Isaac Deneb Castañeda-Alcántara, Arturo Cervantes-Trejo.

**Formal analysis:** Isaac Deneb Castañeda-Alcántara, Arturo Cervantes-Trejo.

**Investigation:** Isaac Deneb Castañeda-Alcántara, Arturo Cervantes-Trejo.

**Methodology:** Zachary Fowler, Lina Roa, Tarsicio Uribe-Leitz, Arturo Cervantes-Trejo.

**Project administration:** Arturo Cervantes-Trejo.

**Supervision:** Tarsicio Uribe-Leitz, John G. Meara, Arturo Cervantes-Trejo.

**Validation:** Zachary Fowler.

**Visualization:** Zachary Fowler, Ellie Moeller, Arturo Cervantes-Trejo.

**Writing – original draft:** Zachary Fowler, Ellie Moeller, Lina Roa.

**Writing – review & editing:** Zachary Fowler, Ellie Moeller, Lina Roa, Isaac Deneb Castañeda-Alcántara, Tarsicio Uribe-Leitz, John G. Meara, Arturo Cervantes-Trejo.

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
