## [Decision Letter · Decision Letter 0]

7 Sep 2020

PONE-D-20-16750

Projected demand for hospital services in the Mexico City Metropolitan Area during the first wave of the COVID-19 pandemic

PLOS ONE

Dear Dr. Fowler,

Thank you for submitting your manuscript to PLOS ONE. After careful consideration, we feel that it has merit but does not fully meet PLOS ONE’s publication criteria as it currently stands. Therefore, we invite you to submit a revised version of the manuscript that addresses the points raised during the review process.

The study design is quite intersting, but major issues can be raised. The rational should be better described in the introduction section, taking into account recent papers (which should be cited and discussed). We suggest that each scenario should be explained in details. 

We look forward to receiving your revised manuscript.

Kind regards,

Chiara Lazzeri

Academic Editor

PLOS ONE

Journal Requirements:

Reviewers' comments:

Reviewer's Responses to Questions

**Comments to the Author**

1. Is the manuscript technically sound, and do the data support the conclusions?

Reviewer #1: Partly

2. Has the statistical analysis been performed appropriately and rigorously? 

Reviewer #1: Yes

3. Have the authors made all data underlying the findings in their manuscript fully available?

Reviewer #1: Yes

4. Is the manuscript presented in an intelligible fashion and written in standard English?

Reviewer #1: Yes

5. Review Comments to the Author

Reviewer #1: This research describes the potential impacts of COVID-19 in this region and to model possible benefits of mitigation efforts. The COVID-19 Hospital Impact Model for Epidemics was used to estimate the probable evolution of COVID-19 in three scenarios: (i) no social distancing, (ii) social distancing in place at 50% effectiveness, and (iii) social distancing in place at 60% effectiveness. The research topic presented in the present paper is interesting, , and relatively well written. In my opinion, the paper can be published after making some major revisions and some improvements in the presentation of the article, which are as follows:

1. Explain the model parameter in more detail.

2. Title should be modified.

3. Adding some finding in abstract.

4. Rewrite introduction part and adding some recent references in the introductory part. Also add the recent references in the introduction part. For example, DOI: https://doi.org/10.1016/j.chaos.2020.109932

5. if possible then rewrite the conclusion part in bullet form.

6. PLOS authors have the option to publish the peer review history of their article (what does this mean?). If published, this will include your full peer review and any attached files.

Reviewer #1: No

---

## [Author Response · Author response to Decision Letter 0]

21 Oct 2020

EDITOR COMMENT: Thank you for submitting your manuscript to PLOS ONE. After careful consideration, we feel that it has merit but does not fully meet PLOS ONE’s publication criteria as it currently stands. Therefore, we invite you to submit a revised version of the manuscript that addresses the points raised during the review process.

The study design is quite interesting, but major issues can be raised. The rational should be better described in the introduction section, taking into account recent papers (which should be cited and discussed). We suggest that each scenario should be explained in detail. 

RESPONSE: Thank you for reviewing our manuscript and providing valuable feedback and insight. We hope that our revisions have addressed the points raised by the reviewers and appreciate the opportunity to resubmit our manuscript. Our responses to the reviewer comments are listed below:

Reviewer #1: This research describes the potential impacts of COVID-19 in this region and to model possible benefits of mitigation efforts. The COVID-19 Hospital Impact Model for Epidemics was used to estimate the probable evolution of COVID-19 in three scenarios: (i) no social distancing, (ii) social distancing in place at 50% effectiveness, and (iii) social distancing in place at 60% effectiveness. The research topic presented in the present paper is interesting, , and relatively well written. In my opinion, the paper can be published after making some major revisions and some improvements in the presentation of the article, which are as follows:

COMMENT 1. Explain the model parameter in more detail.

RESPONSE: Thank you for this comment. We agree with the reviewer that more details of the model parameters should be given. We have modified our reporting of the methods to include sections on “Model parameters” and “Model inputs”. In “Model parameters” we now describe the variables and the equations used to calculate our findings (revised manuscript with track changes, Word document, page 6, lines 170-195). The parameters used include effective contact rate, inverse mean recovery time, doubling time, and growth rate. These are used to model SIR dynamics (susceptible, infected, recovered).

In “Model inputs” we have provided additional information on the numbers used in our model and our rationale for using them (revised manuscript with track changes, Word document, page 7, lines 206-221). The following inputs are used in the calculations and are better described now in this section: regional population, hospital market share, number of patients currently hospitalized with COVID-19, date of the first hospitalization, social distancing, hospitalization rate, ICU admission rate, mechanical ventilation rate, infectious days, average length of hospital stay, average length of ICU stay, and average length of ventilator use. Inputs were based on data reported in Mexico (regional population, hospital market share, number of patients currently hospitalized with COVID-19, date of the first hospitalization, social distancing), from estimates reported by Penn Medicine hospitals in the United States (average length of hospital stay, average length of ICU stay, and average length of ventilator use) and by Verity et al in Lancet Infectious Diseases (hospitalization rate, ICU admission rate, mechanical ventilation rate). 

We have also provided our rationale for selecting the three mitigation strategies (revised manuscript with track changes, Word document, page 7, lines 200-202). This is based on estimates of mitigation level needed to reverse the epidemic and is consistent with strategies used in other prediction models (Anderson et al 2020, Kissler et al 2020, Bhat et al 2020).

We have added an explanation for our inclusion of the entire metropolitan area in our study, rather than limiting it to Mexico City (revised manuscript with track changes, Word document, page 17, lines 422-425). This now reads as follows (additions are underlined): 

Ultimately, this may enable careful resource utilization and swift public health responses that protect the entire MCMA, for which data reporting and public health management are often approached individually by each of the three states within their respective administrative boundaries. Analyses and prediction models that include all municipalities in the metropolitan area can lead to more comprehensive and efficient policies.

COMMENT 2. Title should be modified.

RESPONSE: Thank you for this suggestion. We have modified the title as follows: “Projected impact of COVID-19 mitigation strategies on hospital services in the Mexico City Metropolitan Area.” This title better reflects our aim to compare the impact of various mitigation strategies on hospital service demand (original title: “Projected demand for hospital services in the Mexico City Metropolitan Area during the first wave of the COVID-19 pandemic”).

COMMENT 3. Adding some finding in abstract.

RESPONSE: We agree that it would be important to include more of our findings in the abstract. We have revised the abstract to include additional results from our study. We now report the impact of social distancing on daily hospital admissions, peak intensive care unit occupancy, and peak ventilator demand in the abstract (revised manuscript with track changes, Word document, page 2, lines 49-52).

COMMENT 4. Rewrite introduction part and adding some recent references in the introductory part. Also add the recent references in the introduction part. For example, DOI: https://doi.org/10.1016/j.chaos.2020.109932

RESPONSE: We have revised the introduction to include updated information about the COVID-19 burden in Mexico. This includes new data on confirmed COVID-19 cases in Mexico and globally, positivity rates, case fatality rates, requirements for mechanical ventilation, and excess mortality in Mexico City (revised manuscript with track changes, Word document, page 3, lines 72-98). Information was obtained from recently published academic papers as well as Ministry of Health documents. We have also provided an overview of the types of COVID-19 predictive models that are now in use, including the suggested paper (revised manuscript with track changes, Word document, page 4, lines 115-133). We feel that this will provide readers with a better understanding of the tools available to health policy makers and the need to apply them in Mexico. We have also now referenced in the discussion other articles that describe predictions using this model in other parts of the world (revised manuscript with track changes, Word document, page 17, lines 440-441).

COMMENT 5. if possible then rewrite the conclusion part in bullet form.

RESPONSE: Thank you for this suggestion. We have rewritten the conclusion using bullet points to better organize our message (revised manuscript with track changes, Word document, page 18, lines 464-479).

---

## [Editor Report · Decision Letter 1]

26 Oct 2020

Projected impact of COVID-19 mitigation strategies on hospital services in the Mexico City Metropolitan Area

PONE-D-20-16750R1

Dear Dr. Fowler,

We’re pleased to inform you that your manuscript has been judged scientifically suitable for publication and will be formally accepted for publication once it meets all outstanding technical requirements.

Kind regards,

Chiara Lazzeri

Academic Editor

PLOS ONE
---

## [Editor Report · Acceptance letter]

30 Oct 2020

PONE-D-20-16750R1 

Projected impact of COVID-19 mitigation strategies on hospital services in the Mexico City Metropolitan Area 

Dear Dr. Fowler:

I'm pleased to inform you that your manuscript has been deemed suitable for publication in PLOS ONE. Congratulations! Your manuscript is now with our production department. 

Kind regards, 

on behalf of

Dr. Chiara Lazzeri 

Academic Editor

PLOS ONE